## RESEARCH ARTICLE

# Enhancing knowledge of hypertension among general practitioners in Pakistan through a Train-The-Trainer initiative

Tariq Ashraf[1]*, Rafat Sultana[1], Musa Karim[2], Kanwal Fatima Aamir[2], Mustajab Mujtaba[3], Shoukat Memon[4], Deedar Hussain Gajju[5], Abdul Qadir Bhutto[6], Umair Arif[7], Hassan Irshad Bajwa[8], Naveed Shehzad[9], Haji Maqsood Mehmood[10], Ijaz Ul Hassan[11], Syed Gulzar Ul Hassan[12], Waheed Ashraf[13], Muhammad Saleem[14], Naeem Tariq[15], Muhammad Niaz Khan[16], Khalid Naseem Khan[17], Muhammad Farooq Saeed Khawaja[18], Naveed Hussain[19], Shahid Hussain Memon[20], Imran Ahmed Kazmi[21], Muhamamd Shahzad Azeem[22], Muhammad Akram Asi[8], Khalid Razaq Malik[23], Shahzad Aslam[24], Muhammad Amir Sohail[25], Arshad Ali Shah[3], Maha Zainab Zia Yaqub[26], Syed Khubaib[27], Hasan Imam[28], Ghulam Fareed[29], Rehan Riaz[30]

1 Karachi Institute of Heart Disease (KIHD), Karachi, Pakistan, 2 National Institute of Cardiovascular Disease (NICVD), Karachi, Pakistan, 3 Civil Hospital, Karachi, Pakistan, 4 Red Crescent Institute of Cardiology, Karachi, Pakistan, 5 Mohammad Medical College, Mirpurkhas, Pakistan, 6 Gambat Institute of Medical Science, Gambat, Pakistan, 7 Bahawal Victoria Hospital, Bahawalpur, Pakistan, 8 THQ Hospital, Chichawatni, Pakistan, 9 Central Park Teaching Hospital, Lahore, Pakistan, 10 Chaudhary Hospital, Gujranwala, Pakistan, 11 Mujahid Hospital, Faisalabad, Pakistan, 12 DHQ Hospital, Sargodha, Pakistan, 13 Allama Iqbal Memorial Teaching Hospital, Sialkot, Pakistan, 14 Jhelum Cardiac Centre, Jhelum, Pakistan, 15 Cardiac Hospital, Muzaffarabad, Pakistan, 16 Hayatabad Medical Complex, Peshawar, Pakistan, 17 Khalifa Gul Nawaz Hospital, Bannu, Pakistan, 18 Wapda Hospital, Quetta, Pakistan, 19 DHQ Hospital, Bhimber, Pakistan, 20 Liaquat University of Medical and Health Sciences, Jamshoro, Pakistan, 21 DHQ Hospital, Abbottabad, Pakistan, 22 Mayo Hospital/Bajwa Hospital, Lahore, Pakistan, 23 Sheikh Zayed Hospital, Rahim Yar Khan, Pakistan, 24 Niazi Medical College, Sargodha, Pakistan, 25 Jinnah Hospital, Lahore, Pakistan, 26 University of Texas Medical Branch, Galveston, Texas, United States of America, 27 Saifee Hospital, Karachi, Pakistan, 28 Atia Hospital, Karachi, Pakistan, 29 Peoples University of Medical and Health Sciences, Nawabshah, Pakistan, 30 Faisalabad Institute of Cardiology, Faisalabad, Pakistan

* tariqash45@gmail.com

## Abstract

### Background

Hypertension (HTN) affects over a billion people worldwide, with most cases in low- and middle-income countries (LMICs) where awareness and control remain low. In Pakistan, general practitioners (GPs) are usually the initial contact for hypertensive patients. Through the Train the Trainer (TTT) initiative, a group of consultant cardiologists were trained as master trainers to conduct training for GPs across Pakistan. This study aimed to assess the effectiveness of the TTT initiative regarding knowledge of GPs about the diagnosis and management of HTN.

**Data availability statement:** All relevant data are within the paper and its Supporting information files.

**Funding:** The author(s) received no specific funding for this work.

**Competing interests:** The authors have declared that no competing interests exist.

## Methods

This study included 540 GPs from all over Pakistan. Participants attended HTN training workshops run by Master Trainers under the TTT model and completed a structured online questionnaire in relation to knowledge of HTN before and 1–3 months after training. Knowledge scores were derived from correct responses for 19 items.

## Results

Pre-training GPs demonstrated low HTN knowledge scores with a median of 26.3 [IQR: 15.8–36.8] and 92% scoring less than 50. In contrast, post-training medians increased significantly to 42.1 [IQR: 31.6–63.2], with 38.5% of GPs achieving moderate or high knowledge scores ≥50 (p < 0.001). Overall, gains were observed across all demographic groups; significant improvement was observed among female and younger GPs.

## Conclusions

The TTT initiative effectively improved knowledge in both the diagnosis and management of hypertension among GPs, thus standing to potentially improve the current gaps in HTN care in many LMICs through similar models. Further studies are warranted to document the long-term clinical impact of this kind of training on patient outcomes and hypertension control.

## Introduction

Hypertension is a medical condition defined by chronic elevation of arterial pressure in the systemic arteries above threshold values that have been established [1,2]. Recent estimates show that approximately a third of the world's adult population (>1 billion people) are living with hypertension (HTN), and approximately 50% of these are unaware of their conditions [3]. The global prevalence of HTN among adults surged from 594 million in 1975 to 1.13 billion in 2015 [4], and WHO projections indicate this will rise to 1.56 billion by 2025, representing 29.2% of the global population [5]. Of patients with HTN, roughly 3/4th live in low- and middle-income countries (LMICs); of these, only perhaps 1 in 10 has their blood pressure controlled [3,6–9]. It is estimated that global direct medical costs for the treatment of HTN are approximately US$ 370 billion annually, while healthcare savings from effective management of the condition are projected at roughly $100 billion annually [10]. Aside from its health impact, HTN is now also considered to be an economic problem in LMIC, with direct and indirect economic burdens brought about by the increasing healthcare cost and loss of productivity due to disabilities caused by the disease [8,11].

Epidemiological studies have reported the prevalence of HTN in Pakistan. For instance, one study estimated the rate to be 19.0% from 1990 to 1994 [12] using data from the National Health Survey, and another study specifically conducted in northern rural areas reported a prevalence of 14% in 2001 [13]. A couple of years ago, a

more recent national health survey conducted by Saleem F et al. [14] in 2010 estimated that 33% of adults aged 45 years and 18% of all adults in Pakistan were hypertensive. Interestingly, one-third of the hypertensive clients over the age of 40 years were at risk due to a wide range of diseases. There is a dearth of primary research and updated surveillance data on the prevalence of hypertension in Pakistan. However, estimates from secondary sources, such as the Global Burden of Disease (GBD) study, indicate the age-standardized prevalence rate (per 100,000 population) of hypertensive heart disease is 138.55 for Pakistan in the year 2019 with a 3.79% change between 1990 and 2019 [15].

The survey also clearly showed that out of all the diagnosed hypertensive cases, only half received some form of treatment that resulted in controlled hypertension in just 12.5% of the cases [14]. These statistics underline the urgent need for improved screening, diagnosis, and management strategies in Pakistan, particularly given the associated risks of cardiovascular disease, diabetes mellitus, and chronic kidney disease. Several barriers at the patient, provider, and healthcare system level are identified that are derailing proper control and management of hypertension in LMICs. These include poor or lack of awareness of the condition, inadequate access to hypertension care, poor adherence to the treatment regimen and preventive measures, lack of proper distribution and drug procurement, poor surveillance and screening, lack of trained healthcare staff, and above all lack of proper training and awareness of healthcare providers [11].

HTN is mostly comorbid or precedes many other chronic diseases in Pakistan, such as diabetes mellitus, cardiovascular disease, and chronic kidney disease. Further, it is highly comorbid with behavioral factors, obesity, lack of exercise-and socio-demographic factors like family history [16]. Due to its majorly asymptomatic nature, coupled with the absence of systematic surveillance and screening protocols, HTN mostly goes undetected until its advanced stages, especially in LMICs like ours. Additionally, deviations from recommendations on testing largely contribute to inappropriate diagnosis and management [17]. Generally, patients with the condition or with high-risk features often seek help from general practitioners (GPs) [18]. An improper diagnosis and management by the GPs may result in further aggravation of the situation [17].

The limited availability of continuing medical education and the lack of access to updated guidelines among GPs contribute to poor adherence to best practices. Addressing these gaps requires national-level strategies focused on cost-effective and sustainable interventions, such as structured training programs for primary care providers [19,20]. No attempts are, however, made to increase the GPs' awareness of hypertension, its prevention, and control in the communities in our population. However, the lack of structured training leads to significant gaps in their clinical knowledge and skills, potentially affecting patient outcomes. Traditional continuing medical education (CME) programs may not fully address these deficiencies due to their rigid structures, limited accessibility, and lack of competency-based approaches. To bridge this gap, there is a growing need for flexible, hybrid, and competency-based CME programs that provide tailored learning opportunities for GPs [21]. Consequently, a Train the Trainer (TTT) initiative was launched by the Pakistan Cardiac Society (PCS), the largest cardiology platform in Pakistan. Through this initiative, a group of consultant cardiologists were trained as master trainers to conduct sessions for GPs across Pakistan on the diagnosis and management of HTN.

In the light of above, a series of training for GPs all over Pakistan was planned to enhance GPs' knowledge in the diagnosis and management of hypertension through the TTT approach. The study aimed to assess the effectiveness of the TTT initiative in improving awareness and knowledge regarding the diagnosis and management of HTN.

## Materials and methods

**Study design:** This study employed a quasi-experimental design to assess the pre- and post-training knowledge of GPs regarding the diagnosis and management of hypertension.

**Study setting:** In this study, we included GPs from across Pakistan who attended the hypertension training workshop through a non-probability convenient sampling technique. The study was conducted under the platform of the Pakistan Cardiac Society (PCS) and carried out with the guidance and oversight of its Executive Council. A total of 35 training sessions were conducted by 33 master trainers in 35 locations across Pakistan, including 21 urban and 14 suburban/rural areas, between February 17, 2024, and July 16, 2024 (Central Illustration).

**Ethics:** The study was designed and conducted in compliance with the Declaration of Helsinki and was approved by the Pakistan Medical Association Committee on Ethics (Reference No. BO/056/AMP/09). As data collection was conducted via an online questionnaire, electronic informed consent was obtained from all participants.

The consent form provided detailed information about the research, including its objectives, potential risks and benefits, and participants' rights and responsibilities. Participants were informed that participation was entirely voluntary and that they could withdraw at any time without any repercussions. It was explicitly stated that their decision to participate or decline would not affect their professional standing. Additionally, participants were assured that all personal and medical information would be kept strictly confidential and secured.

**Study participants:** The study participants included GPs, also known as family physicians or general physicians, who were recruited from diverse backgrounds across Pakistan for the HTN training and assessment program. Participants met the inclusion criteria of being between 20 and 75 years old, regardless of gender, currently working as GPs in Pakistan, and having agreed to attend the hypertension training workshop. To minimize potential bias and confounding effects from recent training, GPs who had attended a structured HTN education or training program within the past six months were excluded from the study.

**The Train the Trainers initiative:** The TTT program was a structured training strategy designed for GPs. In the initial phase, 33 volunteer consultant cardiologists from across Pakistan were trained as master trainers for the hypertension awareness and knowledge workshop. Upon completion of their training, each master trainer conducted a GP training workshop at their respective home stations.

An audio-visual module, titled "Hypertension Essentials: A Comprehensive Guide/Course on Hypertension Management," based on the latest hypertension diagnosis and management guidelines, was used for training both master trainers and GPs.

**Assessment**: The baseline and 1–3 months post-workshop knowledge of GPs was assessed using an online multiple-choice structured questionnaire, adapted from a previous study by Chen Q et al. [22] and modified to the regional context. One additional question on salt consumption behavior was included, based on the latest clinical guidelines [23,24].

The study instrument comprised 19 multiple-choice questions related to the diagnosis and management of hypertension and salt consumption. To ensure validity and consistency, the modified questionnaire was piloted with 30 respondents, who were not included in the final study sample.

The online questionnaire was developed using Google Forms, and all enrolled GPs were contacted 1–3 months after the training, with responses recorded using the same instrument and parameters.

**Data collection:** The online questionnaire began with an informed consent section, where participants were informed about the study's purpose, the estimated completion time, and their right to object to the use of collected data for research and publication.

Each participant was allotted 15 minutes before the commencement of the workshop to complete the questionnaire via an online platform. The same questionnaire was used 1–3 months post-training to assess improvements in knowledge.

To protect participant confidentiality, no personally identifiable information was collected beyond consent. The survey data remained accessible only to the research team through a password-protected institutional Google Drive account with restricted access. Additionally, Google's built-in security features, including two-factor authentication, were implemented to safeguard data integrity and prevent unauthorized access.

**Sample size calculation:** The sample size was calculated using the WHO Sample Size Calculator (Version 2.0), based on a previously reported accuracy rate of hypertension prevention knowledge of 49.2%, as cited by Chen Q et al. [21]. Assuming a 10% increase in knowledge post-intervention, with 80% power and a 5% level of significance, the minimum required sample size was estimated to be 388.

Consequently, a total of 547 GPs were enrolled in the hypertension training program through a non-probability convenience sampling approach. However, contact could not be established with seven (7) GPs for the post-training knowledge assessment, resulting in a final analysis of 540 GPs

**Data analysis:** Data analysis was performed using IBM SPSS version 21 and R version 4.3.1. Descriptive statistics such as mean±SD/median and frequency (%) were obtained, and an appropriate McNemar test or Wilcoxon signed rank test was applied to compare the pre and post-correct response rate on each item of the questionnaire. The level of significance was set at ≤ 0.05 throughout the study. The cumulative knowledge score was calculated as 100×percentage correct responses on 19 items and categorized as low, moderate, and high as a score of less than 50, 50–75, and more than 75, respectively.

## Results

The study participants consisted of 540 GPs working across Pakistan. Participants were predominantly male (85.7%) and young with a mean age of 40.7±13.2 years. More than half of the participants had work experience of less than 10 years. The distribution of baseline characteristics of the study participants is given in Table 1.

Poor knowledge regarding diagnosis and management of hypertension was observed with a median score of 26.3 [IQR: 15.8 to 36.8] and 92% (497) were reported to have poor (<50) scores (Fig 1). A significant improvement in overall knowledge score was observed after 30 days of training with a median score of 42.1 [31.6 to 63.2] (p<0.001) and 38.5% (208) were reported to have moderate or high (≥50) scores increased by 8%, highlighting a substantial and meaningful improvement in hypertension knowledge among GPs. Significant improvement was observed in all knowledge parameters (Table 2).

A significant improvement was observed in the pre and post-median scores of males (21.1 [15.8 to 36.8] vs. 42.1 [26.3 to 57.9]) and females (31.6 [26.3 to 42.1] vs. 52.6 [47.4 to 78.9]) participants (Fig 2), emphasizing potential differences in knowledge acquisition that may warrant further investigation.

A significant improvement was observed in the pre and post-median score in each of the age groups with median scores of 31.6 [21.1 to 36.8] vs. 57.9 [36.8 to 68.4], 21.1 [15.8 to 31.6] vs. 42.1 [31.6 to 57.9], and 21.1 [10.5 to 31.6] vs. 31.6 [26.3 to 52.6] among participants aged less than 30 years, 31–50 years, and more than 50 years, respectively (Fig 3). Suggesting that early-career physicians may benefit the most from structured training interventions.

Similarly, a significant improvement was observed in the pre and post-median scores in each of the work experience groups with median scores of 26.3 [15.8 to 36.8] vs. 47.4 [31.6 to 68.4], 26.3 [10.5 to 31.6] vs. 36.8 [26.3 to 57.9], 15.8 [10.5 to 26.3] vs. 36.8 [26.3 to 42.1], and 26.3 [10.5 to 36.8] vs. 36.8 [31.6 to 78.9] among participants with work experience of less than 10 years, 10–20 years, 20–30 years, and more than 30 years, respectively (Fig 4).

**Table 1. Distribution of baseline characteristics of the study participants.**

|  | Summary |
|---|---|
| **Total (N)** | **540** |
| **Sex** | |
| Male | 463 (85.7%) |
| Female | 77 (14.3%) |
| **Mean age** | 40.7±13.2 |
| Less than 30 years | 152 (28.1%) |
| 31 to 50 years | 258 (47.8%) |
| More than 50 years | 130 (24.1%) |
| **Work experience** | |
| Less than 10 years | 298 (55.2%) |
| 10 to 20 years | 129 (23.9%) |
| 20 to 30 years | 47 (8.7%) |
| More than 30 years | 66 (12.2%) |

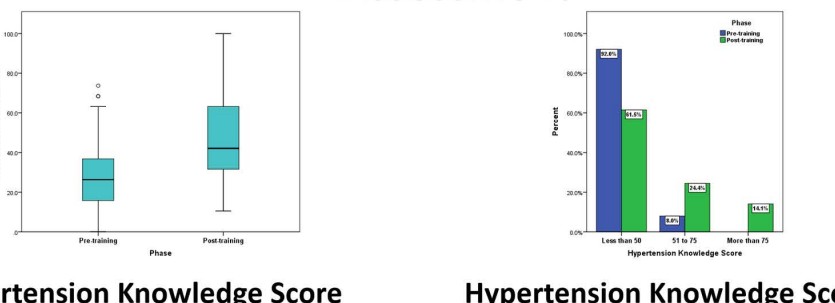

## Phase I
## Train the Trainer

- Training of a total of **33 consultant cardiologist** as the Master Trainers for the hypertension training sessions

## Phase II
## Training Sessions for the General Practitioners

A total of **35** training sessions across **35** locations across Pakistan, **21** urban and **14** sub-urban/rural localities

## Phase III
## Post Training Knowledge Assessment

Hypertension Knowledge Score

Hypertension Knowledge Score

**Fig 1. Effectiveness of the TTT initiative in regards to knowledge of GPs about the diagnosis and management of HTN.**

**Table 2. Distribution of correct response pre- and post-HTN training session.**

| | Phase | | P-value |
|---|---|---|---|
| | **Pre-training** | **Post-training** | |
| **Total (N)** | **540** | **540** | – |
| Q01. Correct diagnosis of hypertension | 350 (64.8%) | 390 (72.2%) | 0.008 |
| Q02. Correct way of detecting patients with hypertension in communities | 23 (4.3%) | 115 (21.3%) | <0.001 |
| Q03. Correct risk stratification with three prognostic risk factors: smoking, obesity, and dyslipidemia | 275 (50.9%) | 435 (80.6%) | <0.001 |
| Q04. Correct descriptions of blood pressure measuring procedure | 47 (8.7%) | 317 (58.7%) | <0.001 |
| Q05. Correct blood pressure control standards | 120 (22.2%) | 271 (50.2%) | <0.001 |
| Q06. Correct non-drug treatment for hypertensive patient | 119 (22%) | 266 (49.3%) | <0.001 |
| Q07. Correct hypertension drug treatment | 125 (23.1%) | 331 (61.3%) | <0.001 |
| Q08. Correct contraindications of angiotensin II receptor blocker (ARB) | 67 (12.4%) | 136 (25.2%) | <0.001 |
| Q09. Correct indications of the dihydropyridine calcium antagonists | 29 (5.4%) | 107 (19.8%) | <0.001 |
| Q10. Correct taboo against the use of diuretics | 184 (34.1%) | 296 (54.8%) | <0.001 |
| Q11. Correct understanding regarding β-blockers | 256 (47.4%) | 326 (60.4%) | <0.001 |
| Q12. Correct inappropriate combination regimens of antihypertensive drugs | 225 (41.7%) | 266 (49.3%) | 0.005 |
| Q13. Correct hypertension-related treatments | 137 (25.4%) | 195 (36.1%) | <0.001 |
| Q14. Correct understanding of populations susceptible to hypertension | 98 (18.1%) | 188 (34.8%) | <0.001 |
| Q15. Correct regarding the management grade of hypertensive patients by general practitioners in the community | 108 (20%) | 209 (38.7%) | <0.001 |
| Q16. Correct referral criteria for newly diagnosed hypertensive patients in the community | 56 (10.4%) | 182 (33.7%) | <0.001 |
| Q17. Correct referral criteria for patients with hypertension who are followed up at community health stations | 167 (30.9%) | 283 (52.4%) | <0.001 |
| Q18. Correct understanding of ambulatory blood pressure monitoring | 209 (38.7%) | 293 (54.3%) | <0.001 |
| Q19. Correct understanding of the recommended daily salt intake for individuals with hypertension | 114 (21.1%) | 230 (42.6%) | <0.001 |
| **Median total score** | 26.3 [15.8 to 36.8] | 42.1 [31.6 to 63.2] | <0.001 |
| Less than 50 | 497 (92%) | 332 (61.5%) | 0.002 |
| 50–75 | 43 (8%) | 132 (24.4%) | |
| More than 75 | 0 (0%) | 76 (14.1%) | |

## Discussion

In Pakistan, well-trained healthcare providers are not sufficient, and almost all GPs are often without access to new evidence-based medical guidelines or resources for CME. Healthcare resources are at a premium in large areas of Pakistan, and thus tailored cost-effective educational interventions are called for. Resulting from the largest cardiology platform in the country, we thus designed a series of training sessions involving consultant cardiologists for GPs across Pakistan to enhance their knowledge regarding the diagnosis and management of HTN.

The result has shown that the TTT approach significantly enhanced the GPs' understanding regarding HTN. Most participants had low levels of knowledge before the intervention, with a median score of 26.3 (IQR: 15.8–36.8), and 92% scoring below 50. Whereas post-training knowledge scores have increased significantly to a median of 42.1 (IQR: 31.6–63.2) and 38.5% achieved moderate to high scores ≥50. Improvement was found across all groups: women and younger GPs with less than 10 years of experience showed the greatest improvement. The low HTN knowledge among GPs at baseline agrees with previously reported findings in LMIC countries, which did not show awareness about HTN among health professionals due to the general scarcity of available training opportunities and resources [28]. Notably, the

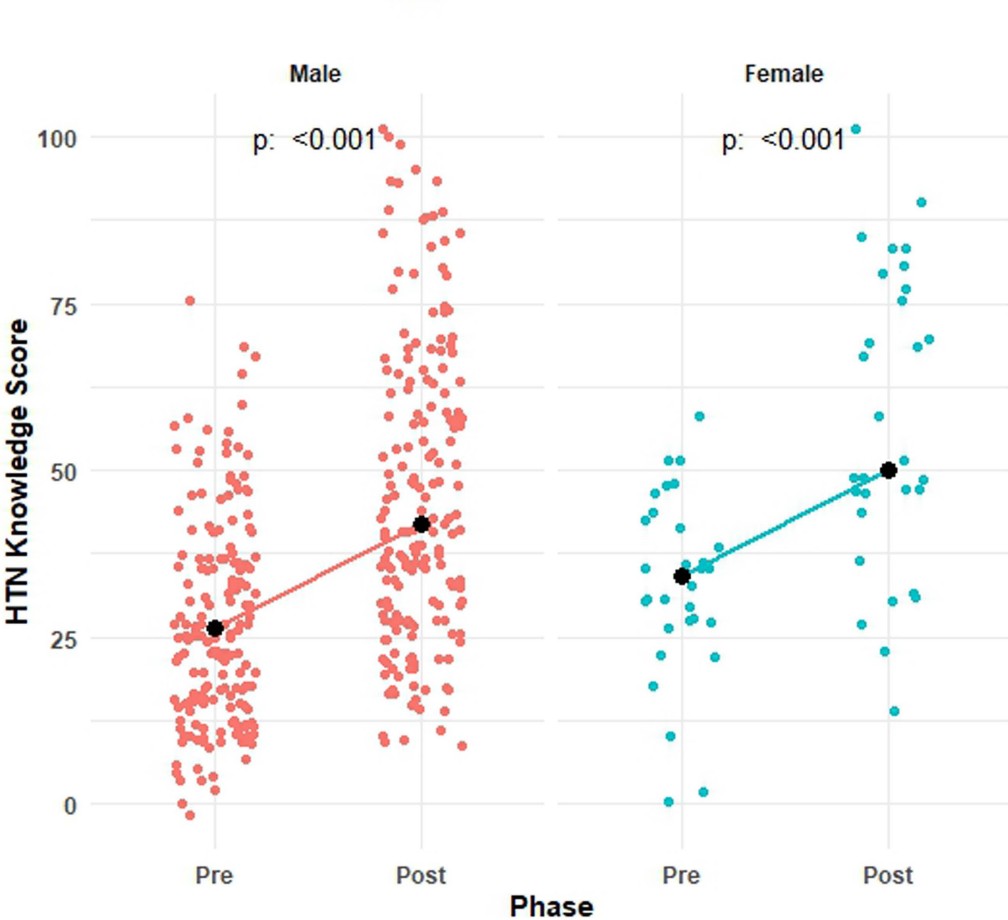

**Fig 2. Distribution and median pre and post-median knowledge score by gender.**

greatest improvements were observed among younger GPs and female participants, reinforcing the potential impact of targeted training interventions. These findings align with previous studies in LMICs, which report similar challenges related to awareness and training opportunities among healthcare providers.

Our study demonstrates that the significant post-training increase in knowledge supports the utility of the TTT programs for enhanced diagnostic skills among GPs. Additional support for this result is demonstrated in other successful TTT models in LMICs, which show that training primary care providers results in improved HTN control rates in their respective communities. Importantly, a parallel study by Mujtaba et al. [25] also investigated the effectiveness of a workshop intervention aimed at improving GPs' knowledge in five cities of Sindh, Pakistan, about the correct methods of BP measurement. It indicated that there was a statistically significant improvement in GPs' knowledge, with mean correct responses increasing from 8.0±2.1 to 14.0±2.5 post-workshop. Dalfó-Pibernat et al. [26] conducted a study to learn the effect of focused training on knowledge regarding ambulatory blood pressure monitoring (ABPM) among nurses and physicians in Spain. The findings indicated that 85.3% of participants reached adequate scores post-training compared to 26.7% pre-training, showing a significant gain in knowledge. The mean global punctuation received an increase of almost 3 points, from 6.3 to 9; 90.5% of the participants scored higher after the intervention. The study conducted by Setia et

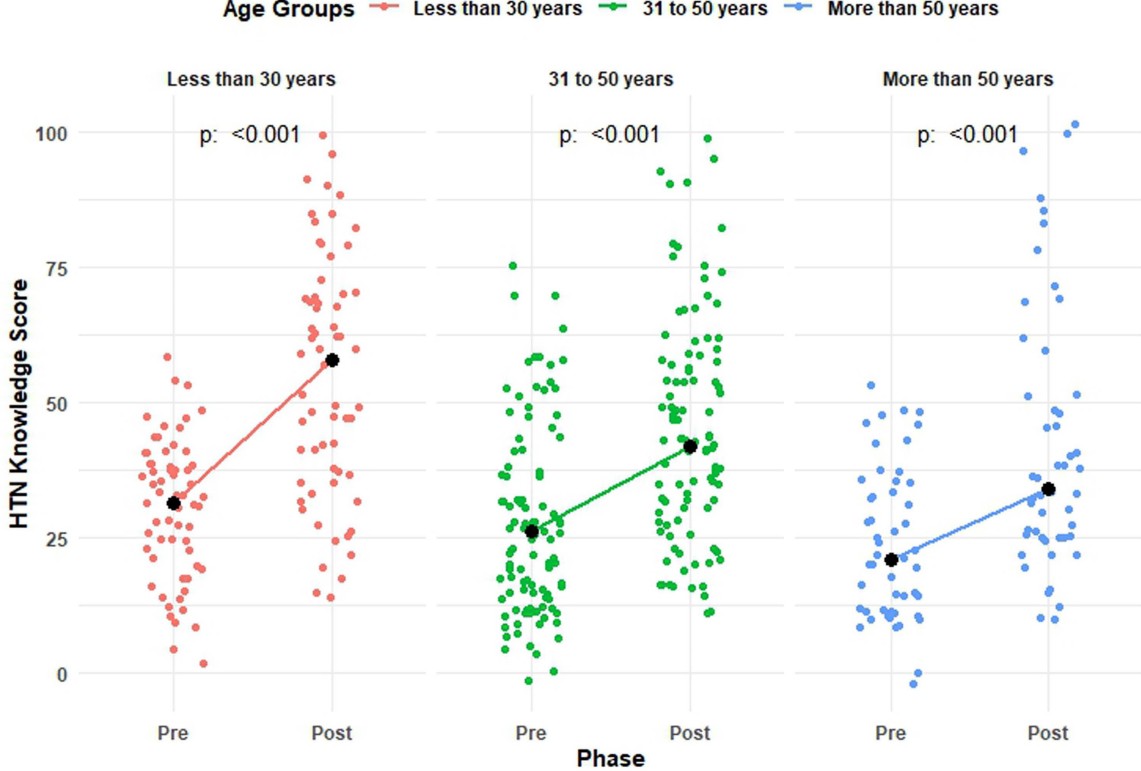

**Fig 3. Distribution and median pre and post-median knowledge score by age.**

al. [27] assessed management practices related to hypertension and blood pressure variability (BPV) among physicians in Singapore. In the same manner, significant gaps in adherence to guidelines and the need for training on BPV were observed. Collectively, these studies underscore the importance of continuous professional development initiatives in optimizing hypertension care.

This questionnaire-based study conducted among 60 physicians from various specialties revealed that while the majority used home blood pressure measurements, only a small fraction of physicians followed the recommended cutoffs of blood pressure for diagnosis. ABPM was most valued by specialists but underutilized, often because of refusals due to cost barriers. The survey indeed found that only 48% of the respondents used the threshold levels of BP recommended by the guidelines to diagnose hypertension, and the pattern of treatments varied among the specialties. For the patients with comorbidities, therapies in combination were preferred. A similar finding reporting a disconnect between guidelines and actual practice was also noted by Green BB et al. [28]. This was reflected in the survey of healthcare professionals in these 10 clinics, wherein most acknowledged the accuracy of manual BP and 24-hour ABPM but generally used clinic BP measurements as a rule for diagnosis. Besides that, few providers followed the recommended threshold of 135/85 mmHg for out-of-office measurements; the use of 140/90 mmHg was common. Although many physicians, physician assistants, and advanced practice nurses expressed a preference for ABPM if readily available, the lack of adherence to guidelines and limited availability of preferred tools suggest barriers to optimal hypertension diagnosis [28].

In the scoping review, Todkar S, et al. [29] present a synthesis of global studies assessing the knowledge, perception, and practice of healthcare professionals regarding BP measurement through the home, ambulatory, automated office, and traditional office BP modalities. Knowledge of home blood pressure measurement (HBPM) and office blood pressure

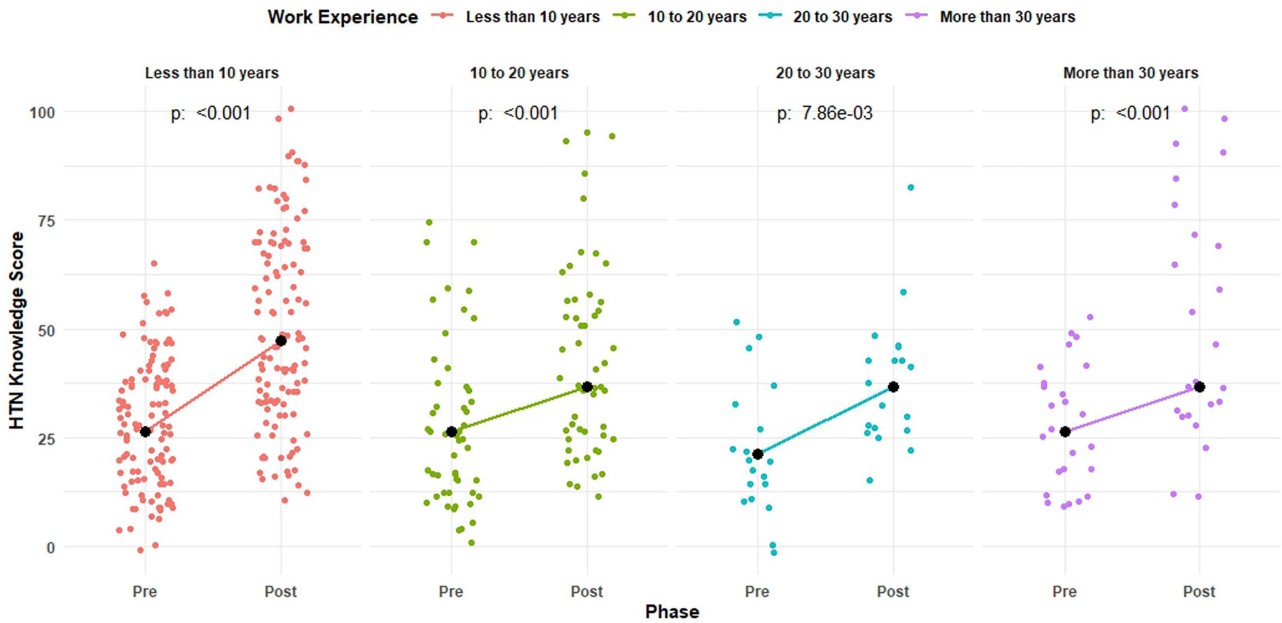

**Fig 4. Distribution and median pre and post-median knowledge score by work experience.**

measurement (OBPM) was suboptimal at 40% unfavorable and 68% unfavorable, respectively. On the other hand, ABPM demonstrated a higher positive knowledge of 86%. Of all these, the overall belief had it that the blood pressure measurement (BPM) methodologies were informative, but the HBPM came close to 80% while the ABPM attained an incredible 93%. However, for the adherence to the hypertension guidelines in practice, the results were all negative for all the BPM methodologies, with the ABPM at 71%, closely followed by the OBPM at 73%, and lastly, the automated blood pressure measurement (AOBP) at 50%. A global scoping review by Todkar et al. [29] further revealed significant discrepancies between knowledge and practice, with adherence to recommended BP measurement methods remaining suboptimal across various settings.

These findings of the study further bring into focus the fact that there is a dire need to have regular targeted training among GPs, as they are usually the first contact for hypertensive patients. Better knowledge of the diagnosis and management of HTN among the GPs will lead to early detection and better education of the patients, thus possibly a reduction in healthcare costs due to unmanaged complications of HTN. This would further help in bridging the gap in HTN care in Pakistan, where a considerable number of patients remain undiagnosed and poorly managed due to poor capacity among specialized providers.

This in turn will ultimately determine the longer-term effect this TTT initiative has on actual clinical practices and patient outcomes, such as BP control rates and incidence of HTN-related complications. This kind of training should be extended to other healthcare providers, such as nurses and community health workers, as a way of further reinforcing hypertension management at the level of primary care. Other post-training reinforcement modalities could include follow-up workshops, online resources, and periodic refresher sessions that would help to maintain knowledge gains over time. A wider review might also consider at greater length the place of the TTT programs in medical curricula with respect to continued knowledge acquisition and use.

Limitations: This study has several limitations. First, the quasi-experimental design without a control group prevents direct attribution of knowledge improvements solely to the Train-the-Trainer (TTT) intervention. Second, the study relied

on self-reported assessments, making it susceptible to social desirability bias and self-selection bias, as GPs who chose to participate may have been more motivated or already more knowledgeable. Third, the follow-up period of 1–3 months is relatively short for assessing long-term knowledge retention, and the study did not evaluate whether knowledge gains were sustained over time. Finally, while the study assessed knowledge improvement, it did not measure whether these changes translated into improved clinical practices or better patient outcomes. Future research should incorporate long-term follow-up assessments and explore the impact of knowledge enhancement on clinical decision-making and patient care.

## Conclusion

The TTT approach has been found to be effective in improving the knowledge of GPs for the diagnosis and management of HTN in Pakistan and may indicate that similar models lead to an important role in the improvement of HTN control in LMICs. This might result in early detection, better treatment, and a reduction in complications of the disease by providing competency in dealing with HTN to primary healthcare providers. Continuation and scale-up of the TTT programs are hereby recommended to further these improvements and, eventually, reduce the burden of HTN in such countries as Pakistan.

## Supporting information

**S1 File. Data file in MS Excel Format.**
(XLSX)

## Acknowledgments

The authors would like to acknowledge the support and guidance of the Executive Council of the Pakistan Cardiac Society. The authors further acknowledge Mr. Qazi Haseeb Alam (Business Unit Head, PharmEvo Private Limited) and Mr. Muhammad Ahsan Zia (Group Product Manager, PharmEvo Private Limited) for their efforts and support in conducting training sessions across the country. The authors would like to acknowledge the research officers, Mr. Zahid ur Rehman, Mr. Sheikh Uzair Abdul Latif, Mr. Uzair Ahmed Siddiqui, Mr. Syed Ammar Uddin, Mr. Muhammad Haris Shaikh, Mr. Sajid Hussain, Mr. Muhammad Irfan, Mr. Muhammad Yousaf, Mr. Umair Aziz, Mr. Aamir Razaq, Mr. Raheel Mustafa Khan, Mr. Muhammad Fayyaz, Mr. Faisal Nawaz, Mr. Nauman Naeem, Mr. Adnan Ali, Mr. Sharjeel Babar, Mr. Muhammad Awais, Mr. Muhammad Ali Zamir, Mr. Muhammad Zohaib, Mr. Muhammad Mubeen, Mr. Shahwaiz Khan, Mr. Roidad Khan, Mr. Zafar Ali, Mr. Mudasar Hayat, Mr. Muhammad Junaid, Mr. Syed Shahzaib Hussain, Mr. Yasir, Mr. Azeem Qamar, Mr. Muhammad Shakir, Mr. Shakir Qureshi, Mr. Shakir Shabbir, Mr. Shahzad ul Haq, Mr. Shakeel Shehzad, Mr. Asim Nisar, and Mr. Atif Habib, for their efforts in recruiting general practitioners for the training sessions.

## Author contributions

**Conceptualization:** Tariq Ashraf, Rafat Sultana, Kanwal Fatima Aamir, Maha Zainab Zia Yaqub.

**Data curation:** Tariq Ashraf, Kanwal Fatima Aamir, Mustajab Mujtaba, Shoukat Memon, Deedar Hussain Gajju, Abdul Qadir Bhutto, Umair Arif, Hassan Irshad Bajwa, Naveed Shehzad, Haji Maqsood Mehmood, Ijaz Ul Hassan, Syed Gulzar Ul Hassan, Waheed Ashraf, Muhammad Saleem, Naeem Tariq, Muhammad Niaz Khan, Khalid Naseem Khan, Muhammad Farooq Saeed Khawaja, Naveed Hussain, Shahid Hussain Memon, Imran Ahmed Kazmi, Muhamamd Shahzad Azeem, Muhammad Akram Asi, Khalid Razaq Malik, Shahzad Aslam, Muhammad Amir Sohail, Arshad Ali Shah, Syed Khubaib, Hasan Imam, Ghulam Fareed, Rehan Riaz.

**Formal analysis:** Tariq Ashraf, Rafat Sultana, Musa Karim, Mustajab Mujtaba, Shoukat Memon, Abdul Qadir Bhutto, Umair Arif, Hassan Irshad Bajwa, Naveed Shehzad, Haji Maqsood Mehmood, Shahid Hussain Memon, Muhamamd Shahzad Azeem, Muhammad Amir Sohail, Maha Zainab Zia Yaqub, Rehan Riaz.

**Funding acquisition:** Tariq Ashraf.

**Investigation:** Kanwal Fatima Aamir, Mustajab Mujtaba, Deedar Hussain Gajju, Abdul Qadir Bhutto, Umair Arif, Hassan Irshad Bajwa, Naveed Shehzad, Haji Maqsood Mehmood, Ijaz Ul Hassan, Syed Gulzar Ul Hassan, Waheed Ashraf, Muhammad Saleem, Naeem Tariq, Muhammad Niaz Khan, Khalid Naseem Khan, Muhammad Farooq Saeed Khawaja, Naveed Hussain, Khalid Razaq Malik, Muhammad Amir Sohail, Arshad Ali Shah.

**Methodology:** Tariq Ashraf, Rafat Sultana, Musa Karim, Kanwal Fatima Aamir, Deedar Hussain Gajju, Hassan Irshad Bajwa, Syed Gulzar Ul Hassan, Muhammad Saleem, Naeem Tariq, Khalid Naseem Khan, Muhammad Farooq Saeed Khawaja, Naveed Hussain, Shahid Hussain Memon, Imran Ahmed Kazmi, Muhamamd Shahzad Azeem, Muhammad Akram Asi, Khalid Razaq Malik, Shahzad Aslam, Muhammad Amir Sohail, Arshad Ali Shah, Maha Zainab Zia Yaqub, Syed Khubaib, Hasan Imam, Ghulam Fareed, Rehan Riaz.

**Project administration:** Tariq Ashraf, Musa Karim, Shoukat Memon.

**Software:** Musa Karim.

**Validation:** Rafat Sultana, Ijaz Ul Hassan, Shahid Hussain Memon, Khalid Razaq Malik, Hasan Imam, Ghulam Fareed.

**Visualization:** Muhammad Farooq Saeed Khawaja.

**Writing – original draft:** Musa Karim, Kanwal Fatima Aamir, Mustajab Mujtaba, Shoukat Memon, Deedar Hussain Gajju, Abdul Qadir Bhutto, Umair Arif, Hassan Irshad Bajwa, Naveed Shehzad, Haji Maqsood Mehmood, Ijaz Ul Hassan, Syed Gulzar Ul Hassan, Waheed Ashraf, Muhammad Saleem, Naeem Tariq, Muhammad Niaz Khan, Khalid Naseem Khan, Muhammad Farooq Saeed Khawaja, Naveed Hussain, Shahid Hussain Memon, Imran Ahmed Kazmi, Muhamamd Shahzad Azeem, Muhammad Akram Asi, Khalid Razaq Malik, Shahzad Aslam, Muhammad Amir Sohail, Arshad Ali Shah, Syed Khubaib, Hasan Imam, Ghulam Fareed, Rehan Riaz.

**Writing – review & editing:** Tariq Ashraf, Rafat Sultana, Maha Zainab Zia Yaqub.

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
