## [Decision Letter · Decision Letter 0]

4 Mar 2025

Dear Dr. Ashraf,

Thank you for submitting your manuscript to PLOS ONE. After careful consideration, we feel that it has merit but does not fully meet PLOS ONE’s publication criteria as it currently stands. Therefore, we invite you to submit a revised version of the manuscript that addresses the points raised during the review process.

Please ensure that the manuscript adheres to the PLOS One formatting guidelines, particularly in terms of using subheadings in the abstract and appropriately placing tables and figures as specified in the guidelines. The "Clinical Perspective" section, located after the conclusion, does not follow the PLOS One formatting structure. It would be better to incorporate this information as a summary elsewhere, possibly within the discussion section.

Furthermore, there is substantial room for improvement in the discussion section. Instead of merely reiterating the results, please include more argumentative points from the authors, providing appropriate comparisons and highlighting the future implications of the findings.

We look forward to receiving your revised manuscript.

Kind regards,

Dr Buna Bhandari

Academic Editor

PLOS ONE

4. We note that there is identifying data in the Supporting Information file < S1_File.xlsx>. Due to the inclusion of these potentially identifying data, we have removed this file from your file inventory. Prior to sharing human research participant data, authors should consult with an ethics committee to ensure data are shared in accordance with participant consent and all applicable local laws.

-Location data

Please remove or anonymize all personal information, ensure that the data shared are in accordance with participant consent, and re-upload a fully anonymized data set. Please note that spreadsheet columns with personal information must be removed and not hidden as all hidden columns will appear in the published file.

Additional Editor Comments:

Please ensure that the manuscript adheres to the PLOS One formatting guidelines, particularly in terms of using subheadings in the abstract and appropriately placing tables and figures as specified in the guidelines. The "Clinical Perspective" section, located after the conclusion, does not follow the PLOS One formatting structure. It would be better to incorporate this information as a summary elsewhere, possibly within the discussion section.

Furthermore, there is substantial room for improvement in the discussion section. Instead of merely reiterating the results, please include more argumentative points from the authors, providing appropriate comparisons and highlighting the future implications of the findings.

Reviewers' comments:

Reviewer's Responses to Questions

**Comments to the Author**

1. Is the manuscript technically sound, and do the data support the conclusions?

Reviewer #1: Yes

Reviewer #2: Yes

2. Has the statistical analysis been performed appropriately and rigorously?

Reviewer #1: Yes

Reviewer #2: Yes

3. Have the authors made all data underlying the findings in their manuscript fully available?

Reviewer #1: Yes

Reviewer #2: Yes

4. Is the manuscript presented in an intelligible fashion and written in standard English?

Reviewer #1: Yes

Reviewer #2: Yes

Reviewer #1: The manuscript is suitable for PLOS ONE, as it aligns with the journal's scope of publishing scientifically rigorous research without a focus on novelty. However, to improve its chances of acceptance, the authors should consider the following revisions:

Major Revisions:

Clarity & Conciseness:

The manuscript is overly wordy in several sections, especially in the Introduction and Discussion. Reducing redundancy and improving clarity will enhance readability.

Example: The background on hypertension prevalence and management in LMICs can be streamlined to focus more on Pakistan-specific challenges.

Study Design & Methodology:

The manuscript states a "quasi-experimental design," but does not specify the control group or if there was a comparison with a non-trained group. Clarify this aspect.

Explain how GPs were recruited for training—was it voluntary, randomized, or based on a specific criterion?

Data Analysis & Interpretation:

The statistical methods are generally appropriate but require more details. Report effect sizes or confidence intervals to strengthen claims of improvement.

While p-values are provided, consider discussing the clinical relevance of observed improvements.

Limitations Section:

The limitations should explicitly mention the potential biases due to self-selection and self-reported assessments.

Discuss the lack of long-term follow-up—while knowledge improved, was there any evidence of sustained application in clinical practice?

Figures & Tables:

The manuscript references multiple figures and tables, but their descriptions could be more precise. Ensure that each figure clearly supports the findings without unnecessary redundancy.

Minor Revisions:

Grammar & Formatting:

Several grammatical errors and awkward phrasings need revision. Example: "Study participants were consisted of 540 general practitioner working across Pakistan" should be "The study included 540 general practitioners working across Pakistan."

Ensure consistency in terminology—sometimes "GPs" is used, other times "general practitioners."

Ethics & Funding Clarifications:

The ethics statement mentions approval from the Pakistan Medical Association, but ensure it meets PLOS ONE's guidelines (e.g., include participant consent procedures clearly).

The funding statement currently states “sponsored by PharmEvo Private Limited.” Clearly outline if the sponsor had any role in study design, execution, or manuscript preparation.

Final Verdict:

The study has merit and is suitable for PLOS ONE, but it requires revisions to improve clarity, methodology description, and interpretation of results. Making these changes will strengthen the manuscript's chances of acceptance.

Reviewer #2: Dear Authors,

I read with deep interest your manuscript on the Train The Trainer (TTT) Initiative to Enhance Hypertension Initiatives in Pakistan. I really found the manuscript interesting and insightful, and believe the topic is of significant public health importance.

I do wish to make some suggestions aimed solely at augmenting the quality of the manuscript.

Please find below:

1. Introduction:

While the discussion on the public health importance of hypertension and its economic impact is comprehensive, incorporating a brief exploration of general practitioners’ perspectives—such as their confidence in diagnosing and managing hypertension—could further enrich this section. Including additional literature from Pakistan, for instance, the manuscript by Farazdaq et al. (see link below), may help underscore the existing gaps in clinical knowledge and the willingness of GPs to enhance their skills through initiatives like the TTT program.

https://journals.lww.com/jfmpc/fulltext/2022/12000/Needs_assessment_of_general_practitioners_in.18.aspx

2. Statistical Analysis:

May the authors elaborate further on which statistical software or tool (e.g. Epidata) was used to calculate the sample size?

3. Data Security:

Given the online and electronic nature of the survey instrument, what specific measures did the authors take to ensure all GPs who participated in the survey's data remained secure? A brief section outlining data management and security measures in the manuscript will be very helpful.

4. Discussion:

The studies cited regarding the prevalence of hypertension in Pakistan are over seven years old, hence it would be helpful if the authors could consider referencing the latest Global Burden of Disease (GBD) study on hypertension, which provides updated statistics relevant to Pakistan. The link is below:

https://www.thelancet.com/journals/eclinm/article/PIIS2589-5370(23)00211-0/fulltext

5. Funding:

Was there a specific grant provided by PharmEvo for the training session? If so, can a specific grant reference number be provided and mentioned in the Funding section of the manuscript?

6. Survey Completion:

Were there any incomplete online survey forms filled by the general practitioners for the study? And if so, how did the authors treat the incomplete online survey forms in their analysis?

7. Instrument Validation:

Since the authors used a modified version of the instrument by Chen Q et al edited for local contexts, did they do a pilot to ensure accurate administration and distribution of the instrument and it's suitability for local contexts? And did they include any results in the final analysis?

8. Language and Grammar:

There are some minor grammatical errors in the sentence structure of the manuscript. For example, the phrase “training of 33 volunteer consultant cardiologist” should be revised to “training of 33 volunteer consultant cardiologists.” in line 150 of the manuscript. A second proofread by a native English speaker, especially of the Methodology section, will help correct these minor lapses.

I once again commend the authors on all their hard work and I look forward to reading more of your work in the future.

**Do you want your identity to be public for this peer review?** For information about this choice, including consent withdrawal, please see our Privacy Policy

Reviewer #1: No

Reviewer #2: **Yes: ** Jaleed Ahmed Gilani

---

## [Author Response · Author response to Decision Letter 1]

16 Mar 2025

Dear Editor,

Thank you for sharing valuable feedback of reviewers regarding our submission to the PLOS ONE titled “Enhancing Hypertension Knowledge among General Practitioners in Pakistan through Train the Trainer Initiative”. Manuscript ID: [PONE-D-24-59868] - [EMID:e69cd29e62458246]

We have revised the manuscript as per the specific comments as listed below. We believe these changes have strengthened the quality of our manuscript and that you will find it suitable for publication in the Journal.

Comment: When submitting your revision, we need you to address these additional requirements.

Response: Title page and manuscript formatting is updated as per the PLOS ONE's style requirements

Comment: 2. PLOS requires an ORCID iD for the corresponding author in Editorial Manager on papers submitted after December 6th, 2016. Please ensure that you have an ORCID iD and that it is validated in Editorial Manager. To do this, go to ‘Update my Information’ (in the upper left-hand corner of the main menu), and click on the Fetch/Validate link next to the ORCID field. This will take you to the ORCID site and allow you to create a new iD or authenticate a pre-existing iD in Editorial Manager.

Response: The ORCID iD for the corresponding author is now linked in the editorial manager

Comment: 3. Your ethics statement should only appear in the Methods section of your manuscript. If your ethics statement is written in any section besides the Methods, please delete it from any other section.

Response: Ethics statement is removed from elsewhere

Comment: 4. We note that there is identifying data in the Supporting Information file < S1_File.xlsx>. Due to the inclusion of these potentially identifying data, we have removed this file from your file inventory. Prior to sharing human research participant data, authors should consult with an ethics committee to ensure data are shared in accordance with participant consent and all applicable local laws.

-Location data

Please remove or anonymize all personal information, ensure that the data shared are in accordance with participant consent, and re-upload a fully anonymized data set. Please note that spreadsheet columns with personal information must be removed and not hidden as all hidden columns will appear in the published file.

Response: The data file is reviewed and efforts are made to omit any identity information

Additional Editor Comments:

Comment: Please ensure that the manuscript adheres to the PLOS One formatting guidelines, particularly in terms of using subheadings in the abstract and appropriately placing tables and figures as specified in the guidelines. The "Clinical Perspective" section, located after the conclusion, does not follow the PLOS One formatting structure. It would be better to incorporate this information as a summary elsewhere, possibly within the discussion section.

Response: The abstract is formatted in according with PLOS One formatting structure and the "Clinical Perspective" section has been omitted.

Comment: Furthermore, there is substantial room for improvement in the discussion section. Instead of merely reiterating the results, please include more argumentative points from the authors, providing appropriate comparisons and highlighting the future implications of the findings.

Response: As per the suggestions, the discussions were elaborated where ever needed.

5. Review Comments to the Author

Reviewer #1: The manuscript is suitable for PLOS ONE, as it aligns with the journal's scope of publishing scientifically rigorous research without a focus on novelty. However, to improve its chances of acceptance, the authors should consider the following revisions:

Major Revisions:

Comment: Clarity & Conciseness:

The manuscript is overly wordy in several sections, especially in the Introduction and Discussion. Reducing redundancy and improving clarity will enhance readability.

Example: The background on hypertension prevalence and management in LMICs can be streamlined to focus more on Pakistan-specific challenges.

Response: Thank you for the valuable feedback, in accordance with the suggestions we have modified introduction and discussion section where ever needed.

Comment: Study Design & Methodology:

The manuscript states a "quasi-experimental design," but does not specify the control group or if there was a comparison with a non-trained group. Clarify this aspect.

Explain how GPs were recruited for training—was it voluntary, randomized, or based on a specific criterion?

Response: Thank you for the valuable feedback, there was not an explicit control group, but the pre-training knowledge level of the participants with the 1-3 months post knowledge level of same participants. Participants were selected based on non-probability convenient sampling methods as mentioned in the methods section.

Comment: Data Analysis & Interpretation:

The statistical methods are generally appropriate but require more details. Report effect sizes or confidence intervals to strengthen claims of improvement.

While p-values are provided, consider discussing the clinical relevance of observed improvements.

Response: Thank you for the suggestions, we have modified the content

Comment: Limitations Section:

The limitations should explicitly mention the potential biases due to self-selection and self-reported assessments.

Discuss the lack of long-term follow-up—while knowledge improved, was there any evidence of sustained application in clinical practice?

Response: Thank you for the suggestions, we have modified limitation section

Comment: Figures & Tables:

The manuscript references multiple figures and tables, but their descriptions could be more precise. Ensure that each figure clearly supports the findings without unnecessary redundancy.

Response: Thank you for the suggestions, we attempted to modify details of figures and table wherever necessary but redundancy is avoided.

Minor Revisions:

Grammar & Formatting:

Comment: Several grammatical errors and awkward phrasings need revision. Example: "Study participants were consisted of 540 general practitioner working across Pakistan" should be "The study included 540 general practitioners working across Pakistan."

Response: Thank you for the suggestions, we have reviewed content for grammatical errors.

Comment: Ensure consistency in terminology—sometimes "GPs" is used, other times "general practitioners."

Response: Thank you for the suggestions, we have updated terminology.

Ethics & Funding Clarifications:

Comment: The ethics statement mentions approval from the Pakistan Medical Association, but ensure it meets PLOS ONE's guidelines (e.g., include participant consent procedures clearly).

The funding statement currently states “sponsored by PharmEvo Private Limited.” Clearly outline if the sponsor had any role in study design, execution, or manuscript preparation.

Response: Thank you for the valuable feedback, consent details and sponsor role are updated.

Reviewer #2: Dear Authors,

I read with deep interest your manuscript on the Train The Trainer (TTT) Initiative to Enhance Hypertension Initiatives in Pakistan. I really found the manuscript interesting and insightful, and believe the topic is of significant public health importance.

I do wish to make some suggestions aimed solely at augmenting the quality of the manuscript.

Please find below:

Comment: 1. Introduction:

While the discussion on the public health importance of hypertension and its economic impact is comprehensive, incorporating a brief exploration of general practitioners’ perspectives—such as their confidence in diagnosing and managing hypertension—could further enrich this section. Including additional literature from Pakistan, for instance, the manuscript by Farazdaq et al. (see link below), may help underscore the existing gaps in clinical knowledge and the willingness of GPs to enhance their skills through initiatives like the TTT program.

https://journals.lww.com/jfmpc/fulltext/2022/12000/Needs_assessment_of_general_practitioners_in.18.aspx

Response: Thank you for the valuable feedback, we have incorporated the suggested study in the introduction section

Comment: 2. Statistical Analysis:

May the authors elaborate further on which statistical software or tool (e.g. Epidata) was used to calculate the sample size?

Response: Thank you for the valuable feedback, we have added software details for sample size calculation

3. Data Security:

Comment: Given the online and electronic nature of the survey instrument, what specific measures did the authors take to ensure all GPs who participated in the survey's data remained secure? A brief section outlining data management and security measures in the manuscript will be very helpful.

Response: Thank you for the valuable feedback, we have added details regarding security measures.

Comment: 4. Discussion:

The studies cited regarding the prevalence of hypertension in Pakistan are over seven years old, hence it would be helpful if the authors could consider referencing the latest Global Burden of Disease (GBD) study on hypertension, which provides updated statistics relevant to Pakistan. The link is below:

https://www.thelancet.com/journals/eclinm/article/PIIS2589-5370(23)00211-0/fulltext

Response: Thank you for the valuable feedback, we have incorporated the suggested study in the discussion section

Comment: 5. Funding:

Was there a specific grant provided by PharmEvo for the training session? If so, can a specific grant reference number be provided and mentioned in the Funding section of the manuscript?

Response: Thank you for the suggestions, this this was not a format grant hence no grant reference number to provide.

Comment: 6. Survey Completion:

Were there any incomplete online survey forms filled by the general practitioners for the study? And if so, how did the authors treat the incomplete online survey forms in their analysis?

Response: Thank you for the suggestions, we have made all the questions mandatory on the survey form, hence there were no missing response, however, the contact could not be established for follow-up assessment with 7 GPs who were excluded from the final analysis. This has been mentioned in the analysis section.

Comment: 7. Instrument Validation:

Since the authors used a modified version of the instrument by Chen Q et al edited for local contexts, did they do a pilot to ensure accurate administration and distribution of the instrument and it's suitability for local contexts? And did they include any results in the final analysis?

Response: Thank you for the valuable feedback, yes the modified questionnaire was tested in a pilot phase of 30 respondents, details are added to the methods seciotn.

Comment: 8. Language and Grammar:

There are some minor grammatical errors in the sentence structure of the manuscript. For example, the phrase “training of 33 volunteer consultant cardiologist” should be revised to “training of 33 volunteer consultant cardiologists.” in line 150 of the manuscript. A second proofread by a native English speaker, especially of the Methodology section, will help correct these minor lapses.

Response: Thank you for the suggestions, we have reviewed content for grammatical errors.

---

## [Decision Letter · Decision Letter 1]

22 Apr 2025

Dear Dr. Ashraf,

Thank you for submitting your manuscript to PLOS ONE. After careful consideration, we feel that it has merit but does not fully meet PLOS ONE’s publication criteria as it currently stands. Therefore, we invite you to submit a revised version of the manuscript that addresses the points raised during the review process.

We look forward to receiving your revised manuscript.

Kind regards,

Dr Buna Bhandari

Academic Editor

PLOS ONE

Journal Requirements:

Reviewers' comments:

Reviewer's Responses to Questions

**Comments to the Author**

Reviewer #2: (No Response)

2. Is the manuscript technically sound, and do the data support the conclusions?

Reviewer #2: Yes

3. Has the statistical analysis been performed appropriately and rigorously?

Reviewer #2: Yes

4. Have the authors made all data underlying the findings in their manuscript fully available?

Reviewer #2: Yes

5. Is the manuscript presented in an intelligible fashion and written in standard English?

Reviewer #2: Yes

Reviewer #2: Dear Authors,

Thank you for sharing your revised submission and for incorporating many of the suggestions to improve the manuscript.

Overall, the work is of good quality; however, several issues—particularly in the Discussion section—still require your attention. Please consider the following detailed suggestions:

1. Ethics Statement and Financial Disclosure

 There is an inconsistency between the manuscript and the information submitted in the portal regarding the Ethics Statement and Financial Disclosure (specifically regarding funding from PharmEvo Private Limited).

 Although the manuscript has been updated per the reviewer’s suggestion in the Methods section and the Declarations section, the corresponding section on the journal’s submission portal under Financial Disclosure and Ethics Statement in the Additional Information section must also be updated accordingly by the submitting author to ensure consistency throughout the submission.

2. Availability of Data and Materials

 The manuscript states in the Declarations section that data will be available upon reasonable request, whereas the Data Availability Statement in the section under Additional Information in the journal submission portal indicates that the data are fully available without restriction as a supplementary file. Please reconcile these differences to ensure consistency between the two sources.

3. Introduction Section

While the Introduction provides a thorough discussion of hypertension, it currently suffers from excessive length and redundancy:

 Approximately two paragraphs are dedicated to discussing the prevalence of hypertension, its economic burden, and the consequences of uncontrolled hypertension. Although these details are important, they may overwhelm busy general physicians and PLOS One readership who are not necessarily specialists in the field and hence, should be trimmed down.

 The Introduction should briefly review existing literature—particularly on Train-the-Trainer initiatives for hypertension management globally—highlight the gaps in the literature, and clearly state how your study addresses these gaps (if any).

 The final paragraph appears abruptly without a clear connection to the preceding content, if a brief review of the current literature on the topic especially on Train-The-Trainer initiatives for hypertension is included, it will flow naturally to this portion of the introduction.

Suggestion: Trim the Introduction to no more than five paragraphs, ensuring that the literature gaps are explicitly discussed.

4. Discussion Section

The Discussion section also requires revision for clarity and coherence. Please consider the following specific points:

a) Citation Needed:

 In lines 226-227, please provide a citation for the statement: “Hypertension is the medical condition defined by chronic elevation of arterial pressure in the systemic arteries above threshold values that have been established.”

b) Statistical Correction and Reference Update:

 The sentence “For instance, one study estimated the rate as 19.1% from 1990 to 1994 [20] using data from the National Health Survey” in line 237 needs two corrections. First, the paper by Tazeen Jafer et al. actually reports the rate as 19.0%. Second, the reference is incorrectly cited as [20]; it should be citation [21] in the reference list. Please amend both aspects accordingly.

c) Citation Placement:

 In the sentence “A couple of years ago, a more recent national health survey conducted in 2010 estimated that 33% of adults aged 45 years and 18% of all adults in Pakistan were hypertensive,” please place the citation for Saleem F et al. (citation [24]) at the beginning of the sentence.

d) Flow and Transition:

 The statement referencing the Global Burden of Disease (GBD) study is inserted abruptly in lines 242-244. Since the following paragraph discusses Saleem F et al. (citation 24), please revise the transition between these two sections to ensure a smoother flow.

e) Abbreviation Introduction:

 The abbreviations ABPM (line 289), HBPM, and OBPM (line 315) are introduced without first providing their full forms. Given the diverse readership of PLOS ONE, please ensure that the full form of each abbreviation is provided at its first occurrence in the manuscript.

f) Structural Revision:

As noted by another reviewer and the editor, the discussion is quite lengthy and meandering. A more focused structure would:

 Start by directly addressing each key finding of the study.

 Compare and contrast these findings with similar studies in the literature.

 Whilst lines 324- 338, highlight the implications of the findings in accordance with the editor's suggestions, it is important, at the same time, to insert appropriate citations with the statements of fact specifically in these two paragraphs.

An example of this would be to provide a citation for the statement "targeted training among GPs, as they are usually the first contact for the hypertensive patient.", and other similar statements in these two paragraphs, which are not inferential in nature but are statements of fact or claim and hence need the appropriate citations from the literature.

 Consider moving the initial two paragraphs—which primarily discuss the clinical definition of hypertension and the epidemiology of hypertension in Pakistan—to the Introduction. Use the Discussion to delve deeply into your study’s findings and their implications relative to previous similar research. This restructuring should help maintain the reader’s attention while providing a concise analysis.

5. Manuscript Title

 Please consider updating the main title to: "Enhancing Knowledge of Hypertension among General Practitioners in Pakistan through a Train-The-Trainer Initiative." This is more grammatically accurate.

 Please also ensure that this title is consistently used throughout the manuscript as well as the submission in the portal and that the capitalization of the words is appropriate.

6. Grammar Check

 It is also suggested that the authors proof read the Discussion and conclusion section of the manuscript as there are still some minor lapses (such as in 353 where the words 'might affect' should be replaced by the words 'might result in') in grammar.

 Either proofreading the entire manuscript one more time by a fluent English speaker or using software like Grammarly may help correct these minor grammatical lapses throughout the manuscript.

Implementing these suggestions will further enhance the quality and clarity of your manuscript and move it closer to publication. The study is well conducted, being of significant public health importance and I look forward to seeing more of your work in the future.

**Do you want your identity to be public for this peer review?** For information about this choice, including consent withdrawal, please see our Privacy Policy

Reviewer #2: **Yes: ** Jaleed Ahmed Gilani

---

## [Author Response · Author response to Decision Letter 2]

4 May 2025

Dear Editor,

Thank you for sharing valuable feedback of reviewers regarding our submission to the PLOS ONE titled “Enhancing Hypertension Knowledge among General Practitioners in Pakistan through Train the Trainer Initiative”. Manuscript ID: [PONE-D-24-59868R1] - [EMID:6454c729025537b3]

We have revised the manuscript as per the specific comments as listed below. We believe these changes have strengthened the quality of our manuscript and that you will find it suitable for publication in the Journal.

Review Comments to the Author

Overall, the work is of good quality; however, several issues—particularly in the Discussion section—still require your attention. Please consider the following detailed suggestions:

Comment 1. Ethics Statement and Financial Disclosure

 There is an inconsistency between the manuscript and the information submitted in the portal regarding the Ethics Statement and Financial Disclosure (specifically regarding funding from PharmEvo Private Limited).

 Although the manuscript has been updated per the reviewer’s suggestion in the Methods section and the Declarations section, the corresponding section on the journal’s submission portal under Financial Disclosure and Ethics Statement in the Additional Information section must also be updated accordingly by the submitting author to ensure consistency throughout the submission.

Response: Thank you for your comments, we have modified Ethics Statement and Financial Disclosure on the portal.

Comment 2. Availability of Data and Materials

 The manuscript states in the Declarations section that data will be available upon reasonable request, whereas the Data Availability Statement in the section under Additional Information in the journal submission portal indicates that the data are fully available without restriction as a supplementary file. Please reconcile these differences to ensure consistency between the two sources.

Response: Thank you for your comments, we have modified data availability statement on the portal.

3. Introduction Section

While the Introduction provides a thorough discussion of hypertension, it currently suffers from excessive length and redundancy:

Comment  Approximately two paragraphs are dedicated to discussing the prevalence of hypertension, its economic burden, and the consequences of uncontrolled hypertension. Although these details are important, they may overwhelm busy general physicians and PLOS One readership who are not necessarily specialists in the field and hence, should be trimmed down.

 The Introduction should briefly review existing literature—particularly on Train-the-Trainer initiatives for hypertension management globally—highlight the gaps in the literature, and clearly state how your study addresses these gaps (if any).

 The final paragraph appears abruptly without a clear connection to the preceding content, if a brief review of the current literature on the topic especially on Train-The-Trainer initiatives for hypertension is included, it will flow naturally to this portion of the introduction.

Suggestion: Trim the Introduction to no more than five paragraphs, ensuring that the literature gaps are explicitly discussed.

Response: Thank you for pointing it out, we have trimmed down the introduction as per the suggestions.

4. Discussion Section

The Discussion section also requires revision for clarity and coherence. Please consider the following specific points:

a) Citation Needed:

 In lines 226-227, please provide a citation for the statement: “Hypertension is the medical condition defined by chronic elevation of arterial pressure in the systemic arteries above threshold values that have been established.”

Response: Thank you for pointing it out, we have added the citation.

b) Statistical Correction and Reference Update:

 The sentence “For instance, one study estimated the rate as 19.1% from 1990 to 1994 [20] using data from the National Health Survey” in line 237 needs two corrections. First, the paper by Tazeen Jafer et al. actually reports the rate as 19.0%. Second, the reference is incorrectly cited as [20]; it should be citation [21] in the reference list. Please amend both aspects accordingly.

Response: Thank you for pointing it out, we have rectified both typographic errors

c) Citation Placement:

 In the sentence “A couple of years ago, a more recent national health survey conducted in 2010 estimated that 33% of adults aged 45 years and 18% of all adults in Pakistan were hypertensive,” please place the citation for Saleem F et al. (citation [24]) at the beginning of the sentence.

Response: Thank you for pointing it out, we have added the citation.

d) Flow and Transition:

 The statement referencing the Global Burden of Disease (GBD) study is inserted abruptly in lines 242-244. Since the following paragraph discusses Saleem F et al. (citation 24), please revise the transition between these two sections to ensure a smoother flow.

Response: Thank you for pointing it out, we have modified the flow of paragraphs.

e) Abbreviation Introduction:

 The abbreviations ABPM (line 289), HBPM, and OBPM (line 315) are introduced without first providing their full forms. Given the diverse readership of PLOS ONE, please ensure that the full form of each abbreviation is provided at its first occurrence in the manuscript.

Response: Thank you for pointing it out, we have added full forms of the abbreviations

f) Structural Revision:

As noted by another reviewer and the editor, the discussion is quite lengthy and meandering. A more focused structure would:

 Start by directly addressing each key finding of the study.

 Compare and contrast these findings with similar studies in the literature.

Response: Thank you for the suggestion, we have moved background information to the introduction section and discussion is now starts with a small background followed by the main findings of the study.

 Whilst lines 324- 338, highlight the implications of the findings in accordance with the editor's suggestions, it is important, at the same time, to insert appropriate citations with the statements of fact specifically in these two paragraphs.

An example of this would be to provide a citation for the statement "targeted training among GPs, as they are usually the first contact for the hypertensive patient.", and other similar statements in these two paragraphs, which are not inferential in nature but are statements of fact or claim and hence need the appropriate citations from the literature.

Response: Thank you for pointing it out, we have added the citation.

 Consider moving the initial two paragraphs—which primarily discuss the clinical definition of hypertension and the epidemiology of hypertension in Pakistan—to the Introduction. Use the Discussion to delve deeply into your study’s findings and their implications relative to previous similar research. This restructuring should help maintain the reader’s attention while providing a concise analysis.

Response: Thank you for the suggestion, we have moved background information to the introduction section as per the suggestions.

5. Manuscript Title

 Please consider updating the main title to: "Enhancing Knowledge of Hypertension among General Practitioners in Pakistan through a Train-The-Trainer Initiative." This is more grammatically accurate.

Response: Thank you for the suggestion, the title is modified as per the suggestions.

 Please also ensure that this title is consistently used throughout the manuscript as well as the submission in the portal and that the capitalization of the words is appropriate.

Response: Thank you for the suggestion, the title is modified in the submission portal too.

6. Grammar Check

 It is also suggested that the authors proof read the Discussion and conclusion section of the manuscript as there are still some minor lapses (such as in 353 where the words 'might affect' should be replaced by the words 'might result in') in grammar.

Response: Thank you for the suggestion, the term is modified as per the suggestions.

 Either proofreading the entire manuscript one more time by a fluent English speaker or using software like Grammarly may help correct these minor grammatical lapses throughout the manuscript.

Response: Thank you for the suggestion, the manuscript has been reviewed for grammatical corrections.

---

## [Decision Letter · Decision Letter 2]

8 Jun 2025

Dear Dr. Ashraf,

Thank you for submitting your manuscript to PLOS ONE. After careful consideration, we feel that it has merit but does not fully meet PLOS ONE’s publication criteria as it currently stands. Therefore, we invite you to submit a revised version of the manuscript that addresses the points raised during the review process.

We look forward to receiving your revised manuscript.

Kind regards,

Buna Bhandari

Academic Editor

PLOS ONE

Journal Requirements:

Additional Editor Comments:

Please address reviewers minor comments

Reviewers' comments:

Reviewer's Responses to Questions

**Comments to the Author**

Reviewer #2: All comments have been addressed

2. Is the manuscript technically sound, and do the data support the conclusions?

Reviewer #2: Yes

3. Has the statistical analysis been performed appropriately and rigorously?

Reviewer #2: Yes

4. Have the authors made all data underlying the findings in their manuscript fully available?

Reviewer #2: Yes

5. Is the manuscript presented in an intelligible fashion and written in standard English?

Reviewer #2: Yes

Reviewer #2: Dear Authors, I read with deep interest your revised manuscript. Thank You for incorporating the suggestions in the paper. I find the work to be fantastic and of significant public health importance. I think the manuscript can now be accepted in its current status. I just had 2 queries which can be addressed in the final proof version upon acceptance by the authors:

1. Were any Research Associates hired to enroll the GPs for this study? Or for any relevant particular purpose of the study? If so, please do acknowledge them with names for this national initiative level study in the final proof version of the manuscript.

2. Whilst inclusion criteria are in the study, was there any exclusion criteria for the GPs in the study? If so, it can be specified in the final proof version of the manuscript. As the authors did exclude GPs of a certain criteria in a past work of theirs: https://www.sciencedirect.com/science/article/pii/S2772628224000335

I congratulate the authors on their fantastic work on the TTT initiative and wish them the best for their academic work and research manuscripts in the future.

**Do you want your identity to be public for this peer review?** For information about this choice, including consent withdrawal, please see our Privacy Policy

Reviewer #2: **Yes: ** Dr. Jaleed Ahmed Gilani

---

## [Author Response · Author response to Decision Letter 3]

27 Jun 2025

Dear Editor,

Thank you for sharing valuable feedback of reviewers regarding our submission to the PLOS ONE titled “Enhancing Knowledge of Hypertension among General Practitioners in Pakistan through a Train-The-Trainer Initiative”. Manuscript ID: [PONE-D-24-59868R2] - [EMID:aebbf8c8c24dc0d2]

We have revised the manuscript as per the specific comments as listed below. Thank you for sharing valuable feedback.

Journal Requirements:

Response: Thank you for the comments, we have rechecked all the references for the completeness.

Additional Editor Comments:

Please address reviewers minor comments

Reviewer #2:

Dear Authors, I read with deep interest your revised manuscript. Thank You for incorporating the suggestions in the paper. I find the work to be fantastic and of significant public health importance. I think the manuscript can now be accepted in its current status. I just had 2 queries which can be addressed in the final proof version upon acceptance by the authors:

Comment 1. Were any Research Associates hired to enroll the GPs for this study? Or for any relevant particular purpose of the study? If so, please do acknowledge them with names for this national initiative level study in the final proof version of the manuscript.

Response: Thank you for the feedback. We have included the acknowledgment for the research officers as suggested.

Comment 2. Whilst inclusion criteria are in the study, was there any exclusion criteria for the GPs in the study? If so, it can be specified in the final proof version of the manuscript. As the authors did exclude GPs of a certain criteria in a past work of theirs: https://www.sciencedirect.com/science/article/pii/S2772628224000335

Response: Thank you for your comment, to minimize potential bias and confounding effects from recent training, GPs who had attended a structured HTN education or training program within the past six months were excluded from the study. Same has been added to the methods section.

---

## [Editor Report · Decision Letter 3]

2 Jul 2025

Enhancing Knowledge of Hypertension among General Practitioners in Pakistan through a Train-The-Trainer Initiative

PONE-D-24-59868R3

Dear Dr. Ashraf,

We’re pleased to inform you that your manuscript has been judged scientifically suitable for publication and will be formally accepted for publication once it meets all outstanding technical requirements.

Kind regards,

Dr Buna Bhandari

Academic Editor

PLOS ONE
---

## [Editor Report · Acceptance letter]

PONE-D-24-59868R3

PLOS ONE

Dear Dr. Ashraf,

I'm pleased to inform you that your manuscript has been deemed suitable for publication in PLOS ONE. Congratulations! Your manuscript is now being handed over to our production team.

Kind regards,

on behalf of

Dr. Buna Bhandari

Academic Editor

PLOS ONE